# A Longitudinal Prospective Study: The Effect of Annual Seasonal Transition and Coaching Influence on Aerobic Capacity and Body Composition in Division I Female Soccer Players

**DOI:** 10.3390/sports8080107

**Published:** 2020-07-30

**Authors:** Troy M. Purdom, Kyle S. Levers, Chase S. McPherson, Jacob Giles, Lindsey Brown

**Affiliations:** 1Department of Kinesiology, North Carolina Agriculture and Technical State University, 1601 East Market Street, Greensboro, NC 27411, USA; 2Department of Exercise and Nutrition Sciences, George Washington University, Washington, DC 20037, USA; klevers@gwu.edu; 3Department of Health, Athletic Training, Recreation, and Kinesiology, Longwood University, Farmville, VA 23909, USA; chasemcpherson14@gmail.com (C.S.M.); jacob.giles@live.longwood.edu (J.G.); brownl12@mymail.vcu.edu (L.B.)

**Keywords:** VO_2max_, periodization, detraining, training adherence, directed training, indirect coaching model, transition period

## Abstract

This study assessed how seasonal transitions and coaching influence affect aerobic capacity (AC) and body composition across the annual training cycle (ATC). Eleven division 1 female soccer players were tested after five predesignated time blocks (B1–B5): post-season 2016 (B1), nine-week transition (B2), spring season (B3), pre-season (B4), and post-season 2017 (B5). Height, weight, and body composition (fat-free mass (FFM)) were measured prior to a standardized 5 min treadmill running and dynamic movement warm up before a maximal AC test. Statistical analysis included a 4 × 5 repeated-measures analysis of variance (ANOVA) (dependent variable × time) with the Fishers Least Significant Difference (LSD) post-hoc test when relevant; data are presented as mean ± standard deviation, effect size (ES), and percent change (%). The statistical analysis revealed that the ATC had a significant main effect on AC and FFM (F_3,4_ 2.81, *p* = 0.001; η^2^ = 0.22). There were significant increases in AC across the transition period (B1–B2) with reduced training volume (∆ + 12.9%, *p* = 0.001; ES = 0.50) while AC and FFM peaked after the spring season with directed concurrent training paired with adequate rest B1–B3 (∆ + 16.4%, *p* < 0.01; ES = 0.81). AC decreased across the pre-season with indirect training (B3–B4) (∆ − 7.0%, *p* = 0.02; ES = 0.50) and remained suppressed without change (*p* > 0.05) across the competitive season (B4–B5). Rest, concurrent training, and directed training positively affected AC, while indirect training and high training loads with little rest negatively affected AC.

## 1. Introduction

Seasonal training stress variation throughout the annual training cycle (ATC) is known to affect aerobic capacity (AC) [1,2] and body composition [1,3,4,5]. Physiological adaptation is the result of exercise stress and then recovery, which varies according to training volume, intensity, frequency, and influenced by seasonal transitions [6]. Cumulatively, training volume, intensity, and frequency make up total training load, which influences adaptation. Training goals to affect adaptation are known to differentiate throughout the ATC with consideration of the competitive and off-season training [1,2,3,7], in addition to the transition period(s) [6,8]. Variation in training loads and goals throughout the ATC are typically divided into training periods and include but are not limited to the pre-competitive, competitive season, transition (detraining), and an off-season training period [5,6]. Training periods are further clarified according to individualized and team sport demands and generally recognized as common practice in collegiate and professional sports. While soccer is a sport that prioritizes AC in combination with repeated short anaerobic sprints to maintain optimal performance [2,3,4,5,9], the literature has yet to agree how the training periods affect AC across the ATC when accounting for team training dynamics, sport demands [1,3,5,7,9], and coaching strategies [10,11], which is further complicated by the lack of female presence in the current literature. 

Training type [6,10] within each training period has been shown to vary across the ATC depending on the time of year [6,12,13,14] in addition to intrinsic and extrinsic factors [6], e.g., weather, training cycle, and competitions, respectively. Coaches balance intrinsic and extrinsic limitations with yearly planning for peak physiological adaptation [3,5,6,12] while attempting to reduce risk of overtraining [6,8,12] during the competitive season. To accommodate these limitations, coaches typically include two variant styles: direct and indirect. Directed coaching uses direct oversight to facilitate interventions and increase adherence [10], while indirect is a passive method absent of direct contact with the coaching staff. Aerobic capacity is highly regarded as a determinant of soccer performance at elite levels [2,15,16,17] and is thought to very according to seasonal variation [2,6,8] despite evidence suggesting otherwise [3,5]. Therefore, to consider the impact of seasonal transition(s) with consideration of coaching influence and more specifically the variation in training periods (planned or not) throughout the ATC is necessary to optimize performance [6,8] and protect athletes from developing overtraining syndrome [18,19,20]. 

Within the ATC, training periods typically have a specified purpose/goal(s). For example, the transition period is defined as a complete cessation or significant reduction in training load [6,8] and can range from 2 to 8 weeks [8]. During a transition period, extended rest has been shown to reduce AC [6,21] by as much as 20% in competitive endurance athletes [8] and with no differences in whole-body FFM [1,15]. The decline in AC throughout the transition period [4,8] as a result of reduced training volume is thought to be attributed (in part) to a reduction in plasma volume, cardiac dimension [8,22], and ventilatory efficiency [8]. While the transition period is suggested to extend beyond 4 wks to recover from metabolic and tissue stress [6], the preparatory “season” and/or pre-competitive period [4,6,14] following the transition can include a sudden increase in training load (typical of the pre-competitive season) [6]. Without appropriate seasonal progression/planning, the sudden increase in training load could have deleterious effects on how the players perform during the competitive season, specifically decreased AC [4,6,23] and increased injury risk [6]. Furthermore, the large training loads the pre-season frequently includes is akin to concurrent training (CT) [6] and incorporates a combination of strength and endurance exercises [1,3,5,23]. The rapid increase in training load coaches employ during the pre-competitive season following the transition period (recovery) [4] paired with large-volume exercise such as CT can perpetuate physiological and psychological stresses. Most often, the broad accumulation of stress can facilitate increases in fatigue prior to the competitive season, leading to an inverse relationship with performance [6]. 

Accumulated training stress experienced during the pre-competitive season can overexert athletes from an overreached state (<2 weeks) [19,23] to an overtrained state (>2 weeks) [4,6] during the competitive season. Overtraining is defined by excessive training stress paired with little rest between competitions and training [24,25], marked by a sustained reduction in performance beyond two weeks, and is typically accompanied by chronic fatigue, respiratory infections, and mood swings [4,6]. Frequent high-intensity competitions, practice, and variable off-season training can lead to accumulated stress, which alters physiological performance (both positive and negative have been observed) [3,4,5]. The continual stress that athletes experience while training and competing can create an imbalance in the homeostatic anabolic and catabolic muscular processes, which can negatively influence performance [26]. Moreover, female athletes are 36% more likely to become overtrained compared to male athletes (26%) [27], further affecting training and competition performance. 

Current research on trained adult soccer players (age ≥ 18) suggests that body composition fluctuates [6] throughout the calendar year, while studies that include collegiate athlete populations indicate that the ATC does not have an effect on body composition [3,5]. Considering body mass is included in relative AC measurement (mL·kg^−1^·min^−1^) [2,8,15,16,17], changes in body composition can influence AC [14,23] measurement and performance. Moreover, the primary oxidative tissue relevant to AC is fat-free mass (FFM) [15]. Therefore, increases in FFM from CT would likely increase AC [4,14,23]. Contrariwise, adipose tissue does not contribute to maximal oxygen consumption and therefore an inverse relationship exists between fat tissue and AC [15,28], further demonstrating the effect of body composition on AC. Therefore, the aim of this study was to evaluate the effect of seasonal transition, training stress, and coaching influence throughout the ATC on AC with consideration of body composition in Division I female soccer players. 

## 2. Materials and Methods

### 2.1. Exercise Design

Using a repeated-measures design, this study investigated the effect of seasonal transition throughout the ATC on AC and body composition. Aerobic capacity (mL·kg^−1^·min^−1^), body weight (BW) in kilograms (kg), body fat percentage (%BF) expressed as a percent of body weight (%), and FFM (kg) were collected across five predesignated seasonal transitions where training focus varied— post-season 2016 (B1), transition (B2), spring season (B3), pre-season (B4), and post-season 2017 (B5)—as shown in Figure 1. Exercise testing occurred at the end of each separate season marked by time blocks (B1, B2, B3, B4, B5). Sample size was determined using an a priori power analysis with an effect size of 0.68 from Miller et al. [3] and a power of 0.98. Potential subjects were excluded if they self-reported or had any diagnosed cardiovascular, metabolic, pulmonary disorders, pregnancy, or had experienced any orthopedic injuries preventing safe testing. This study was conducted according to the Declaration of Helsinki guidelines and all procedures were approved by the University Institutional Review Board. Twenty-two subjects who had completed a full National Collegiate Athletics Association (NCAA) Division 1 (D1) season prior to the study agreed to participate after providing informed written consent. Subjects completed a health history questionnaire and were informed of pre-test guidelines prior to each testing block which consisted of refraining from exercise for a minimum of 24 h, avoiding stimulants/depressants including caffeine for 12 h, and fasting for four hours before all testing blocks. Subject attrition included five subjects who dropped from the study—three due to positional characteristics (goalies), and three due to injury and/or illness (*n* = 11). 

### 2.2. Subjects

Eleven (*n* = 11) Division I female soccer players (mean ± SD: 19.3 ± 1.0 years; 164 ± 6.4 cm; 60.1 ± 5.4 kg; 19.4 ± 3.5% BF, 48.3 ± 4.0 kg FFM, 43.3 ± 3.3 mL/kg/min VO_2max_) were included in the study. All included subjects completed a series of graded exercise tests (GXTs) to assess AC along with body composition over five predesignated time blocks (B1–B5). For each testing block, after confirmation of pre-test guideline adherence subjects completed a urine-based pregnancy test as pregnancy was an exclusion criterion to maximal testing [30]. Throughout the ATC, subjects arrived to the lab for each testing block, where height and weight were measured via an electric standiometer and a scale (Seca Corp., Chino, CA, USA). Body density was measured by a trained researcher using handheld skinfold calipers (Beta Technology, Santa Cruz, CA, USA) and the three site skin fold method: triceps, suprailiac, and thigh [30]. Body composition was then estimated (%BF) using the Brozek conversion equation [30,31]. 

### 2.3. Aerobic Capacity Testing

Prior to each exercise test, subjects were fitted with a heart rate (HR) monitor (Polar Inc., Warminister, PA, USA) and completed a standardized dynamic warm up (high knees, butt kickers, and high bounds) prior to a 5 min self-selected warm up to acclimate to treadmill (Life Fitness Inc., Rosemont, IL, USA) running. Expired gases were measured (ADInstruments Inc., Sydney, Australia) along with HR and RPE at 1 min intervals. Heart rate was measured using a wireless signal integrated into the metabolic cart from a chest strap. Each treadmill GXT protocol was designed to begin with a brisk walk (1.6–1.8 m·s^−1^). Each 1 min stage increased velocity by 0.22–0.44 m·s^−1^ to ensure that volitional fatigue would occur within 8–12 min [32,33]. Standardized criteria for the determination of VO_2max_ along with a 15 s running average as described by Robergs et al. 2010 [34] were used to determine VO_2max_. Each testing block repeated the above procedure across the ATC at the five predesignated time blocks (B1–B5), as shown in Figure 1. 

### 2.4. Specified Training Season Focus

Each testing block followed periodized training periods with pre-determined team training goals implemented by the coaching staff depicted in Figure 1. The transition period (B1–B2) totaled 10 weeks and immediately followed the 2016 competative season. Moreover, the transition period included an indirect training method (no direct coach oversight) with no specified exercise prescription and a NCAA regulated/limited coaching communication period [29]. The 12 week spring season (B2–B3) increased training volume, frequency, and intensity, which included regular bouts of directed (strength coach facilitated) CT (≥3× week) along with regular aerobic and velocity training that had a sport specific focus. Fifty percent of the spring season consisted of an 8 h/week CT period and the remaining 50% 20 h CT weeks. The pre-season (B3–B4) was 15 weeks in length and employed an indirect training method (no direct coach oversight), where subjects were given a periodized CT program with specified aerobic, anaerobic, and resistance training exercises that included frequency, volume, and intensity prescriptions. Throughout the pre-season, subjects participated separately in “summer league play”, with periodic competitions that varied sample wide. The pre-season was also a NCAA-mandated period of limited coach communication [29] and therefore the potential for sample wide variation in training volume, intensity, and frequency was high. Lastly, the competative season (B4–B5) spanned 15 weeks, included a minimum of two NCAA collegiate competitions per week and ~four days of CT training/practice per week.

### 2.5. Statistical Analysis

A 4 × 5 repeated-measures ANOVA was used to analyze dependent variables: AC (mL·kg^−1^·min^−1^), FFM (kg), %BF (%), BM (kg) across five time blocks (B1–B5) using Statistical Package for the Social Science (SPSS), Version 23 (IBM Corp., Armonk, NY, USA). Alpha level of significance was set to (*p* < 0.05) and data are presented as the mean ± SD, percent change (Δ%), and effect size (ES). Percent change was calculated as % Δ = ((final − initial/initial) x 100) and predesignated ES range limits were established as: low effect = 0.20–0.49; medium effect = 0.50–0.79; large effect = 0.80–1.0. When significant differences were observed, the Fisher’s Least Significant Difference (LSD) post-hoc test was used to investigate relevant significant interactions between variables across time. 

## 3. Results

The ANOVA test revealed a significant main effect of time on AC and body composition throughout the ATC (F_3,4_ = 2.81; *p* < 0.001; η^2^ = 0.22; power = 0.99). No differences were found (*p* > 0.05) in BF% across the ATC and therefore it was removed from the analysis. Pairwise comparisons for AC and FFM are presented as the mean ± SD along with %Δ and ES shown in Table 1 and Table 2. Following the nine-week transition period (B2), aerobic capacity significantly increased (+6.41 mL·kg^−1^·min^−1^, *p* = 0.001; CI: 3.17/9.66) without changes in FFM (*p* > 0.05) (Table 1 and Table 2) despite a reduced training load (detraining). Aerobic capacity continued to increase significantly after the spring season training (B3) compared to B1 where AC peaked (+8.47 mL·kg^−1^·min^−1^, *p* = 0.001; CI: 5.61/11.33), but the change was not shown to be significant (*p* > 0.05) compared to B2. Additionally, BM increased from B1 to B3 (+1.66 kg, *p* = 0.03; CI: 0.21/3.01), but is consistent with an increase in FFM during the same time period (+0.98 kg, *p* = 0.02; CI: 0.21/1.74). After B3, there was a significant decrease in AC across the pre-season (B3–B4) training period (−3.65 mL·kg^−1^·min^−1^, *p* = 0.02; CI: −0.68/−6.62) with a reduction in BM (−1.18 kg, *p* = 0.03; CI: −2.21/−0.14). Furthermore, AC decreased across the competitive season from peak (B3–B5) (−4.78 mL·kg^−1^·min^−1^, *p* = 0.01; CI −1.63/−7.93). No significant changes were observed across the competitive season (B4–B5) in any metric despite a 0.84 mL·kg^−1^·min^−1^ decrease in AC with a corresponding increase in FFM (0.56 kg) during the same time period. 

## 4. Discussion

The purpose of this study was to observe the effect of coaching facilitated seasonal variation throughout the ATC on AC with consideration of body composition in Division 1 female soccer players. Currently, there are four studies examining seasonal variation effect which lack a consensus on the longitudinal impact on AC and body composition [1,2,3,5]. Of the four studies, only Caldwell and Peters [1] and Tønnessen et al. [2] reported significant differences in AC and BF%. The reported differences in AC and BF% were based upon moderate and low effect sizes (0.53 and 0.10, respectively) in male professional soccer players that only occurred during the transition period (off-season/detraining period). Our sample, which included D1 female collegiate athletes, further differed from Miller et al. and Pearl et al. [3,5], who observed no differences in AC across the ATC in D1 collegiate athletes. Our results indicate a substantial seasonal transition influence on AC, with the largest effect size of 0.83 occurring from the post-season through the end of the pre-season (B1–B3), representing a >16% increase in AC. Furthermore, our results indicate that seasonal variation throughout the ATC influenced both AC and FFM over the annual training calendar (Table 1 and Table 2; Figure 2), which is likely due to the variation in coaching strategies deployed during each of the ATC periods in D1 female soccer players. 

The transition period following the 2016 competitive season (B1–B2) included an indirect coaching method that included a significant reduction in overall training load and no training prescription for 10 weeks. The training load was significantly reduced alternative to the preceding competitive season which included six training days/week for ~14 weeks. Furthermore, the D1 collegiate soccer transition period has a NCAA regulated (limited) coach communication with athletes [29]. The combined effect of no training prescription and limited communication produced a 13% increase in AC compared with professional soccer players who reported a ~3% reduction [1,2]. In contrast, Miller et al. [3] and Peart et al. [5] reported no differences in AC throughout the ATC in female collegiate soccer players. While the transition period typically includes reductions in cardiovascular performance [8,27], and therefore AC [27], our sample experienced a wide range of responses demonstrated by the large standard deviations in AC and FFM (43.34 ± 6.87 mL·kg^−1^·min^−1^; 48.1 ± 4.09 kg), respectively. The limited coaching direction employed during the transition period is known as an indirect coaching method and can affect a wide variety of responses and adherence from athletes [6,8,10,11]. The differences observed within our population compared with elite athletes is something to note and can be due training prescription management throughout the competitive season. An increase in AC of this magnitude after an extended reduction in training load suggests our population suffered from over training syndrome. The lack of necessary recovery from the large doses of intense exercise inherent to the competitive season are known to limit performance [4,6,12] and perpetuate injury. Nonetheless, throughout the transition period, a significant reduction in training load was shown to increase AC and is suggestive of the importance of rest throughout the ATC.

The spring season (B2–B3) programming impetus focused on CT that included a rapid increase in training load compared to the transition period (B1–B2). Incorporating CT can be problematic for athletic populations because it can lead to sudden and unnecessary increases in accumulated systematic stress [6,20] that can perpetuate overtraining syndrome [4,6,12]. Athletes who incur significant amounts of stress without appropriate rest can develop overtraining syndrome [4,6], which is known to negatively impact AC [4]. However, in our sample, CT was initiated with a reduction in total training/week for 50% of the spring season (B2–B3), while the second portion included 20 h/week (Figure 1). The ramped increase in training load paired with CT produced a peak AC after the spring season (B2–B3) along with a significant increase in FFM (*p* < 0.03), as shown in Figure 2. The responses to CT (increased AC and FFM) observed in this study agree with previous research that adequate amounts of rest were included to recover from the greater training load and concurrent style of training [14,21,23]. Concurrent training has been shown to decrease %BF while increasing FFM and AC if appropriate rest is included [14,20,35], which our population experienced. The directed CT limited to 8 h/week paired relevant stimulus and provided significant rest intervals compared with the competitive season (B4–B5), and therefore proved to be an effective strategy to facilitate peak physiological potential within the ATC.

Following the spring season, a 7% decrease (*p* < 0.02) in AC from peak was shown to occur after the pre-season (B3–B4) without a change in FFM (Figure 2). Typically, the pre-season is meant to focus on performance development [6]. During this timeframe, our population was given a periodized training plan that detailed daily training exercises, volume, and intensity prescriptions to complete without direct contact with the coaching staff. Our findings indicate that the coaching strategy deployed in the pre-season (B3–B4) resulted in negative outcomes despite the attempt to increase and/or maintain peak physiological performance capacity observed in the spring season (B2–B3) (Figure 2). Moreover, the reduction in AC following the spring season (B2–B3) continued to decline through the end of the competitive season (B5) (B3: 51.81 ± 2.67 mL·kg^−1^·min^−1^; B5: 47.03 ± 3.14 mL·kg^−1^·min^−1^; *p* = 0.01) with a moderate effect size (ES = 0.65). The reduction in AC occurred despite a 0.36% non-significant increase in FFM (Figure 2). Lack of variation in AC and body composition throughout the competitive season despite significant training loads (training + competitions) is not novel. However, increased training load paired with little recovery, common to the competitive season [5], may have perpetuated the sustained suppression of AC from peak (B3) that could not be recovered from. This finding suggests that pre-season programing and preparation are instrumental to competing at peak physiological potential. Additionally, the suppression of AC without a change in FFM further demonstrates that AC is independent of FFM, which has been reported in previous literature [1,3,8,15,16].

The decline in AC throughout the pre-season (B3–B4) could have been due to poor training adherence. Athletes that train under the indirect coaching style utilized during the pre-season have been shown to experience a perceived lack of coaching support [10]. The NCAA DI guidelines [29] limit player/coach communication throughout the summer or pre-season (B3–B4) for fall sports (Figure 1), which can create training programming and management barriers. The indirect coaching style and potential for a perceived lack of coaching support can result in poor training adherence [11]. Moreover, physical barriers such as limited access to training facilities as well as perceived barriers, e.g., lack of coaching feedback, and by extension support, are known to limit training motivation and adherence [10,11]. Poor training adherence can negatively affect training volume and frequency, which can ultimately lead to deleterious consequences noted in our sample across the pre-season (B3–B4). More concerning was our population’s lack of ability to regain full physiological potential following the decrease despite large volumes of training/competition during the competitive season (B4–B5). These findings suggest an indirect training program that can create both perceived and physical barriers can have detrimental effects on athletes physiological potential that extend throughout the competitive season. In contrast, directed training/programing during the pre-season, similar to the training style during the spring season (B2–B3), could serve as an optimal training style to maintain and/or increase peak physiological capacity that extends throughout the competitive season if adequate rest is included within the training prescription.

The spring season (B2–B3) elicited the greatest positive physical adaptations and was likely due to direct coaching oversight [11] and CT paired with rest [14,21,23]. During the spring season, our population had regular access to sport-specific coaches, strength and conditioning staff, and athletic trainers. The directed coaching strategies our population received across the spring season align with relevant literature, leading to greater training adherence [10,11] and therefore more consistent and positive training adaptations. This same direct coaching support was not present during the pre-season (B3–B4), which likely led to variations in training adherence [10,11] and therefore negatively impacted training volume and intensity. The lack of direct coaching support our population experienced across the pre-season (B3–B4) led to a −7.0% (*p* = 0.021) decrease in AC despite summer league competitions and self-directed training (Figure 1). The drop from peak in AC across the pre-season decreased the population’s competitive season physiological potential and was unable to recover from the decrease throughout the competitive season (B4–B5). The training load athletes experience throughout the competitive season while lacking recovery can put athletes at risk for unnecessary fatigue [4] (inability to recover from training stress), increased risk of injury [6,36] along with suppression of training adaptations which include AC [1,6,27]. Therefore, the spring season preparatory training represents a pivotal opportunity that can positively or negatively affect competitive season performance with little opportunity to reverse unintended consequences.

Current literature supports the notion that planned peak physiological status is affected by the training volume and frequency that occurs during the off-season [pre-season] [6,8]. Our population maintained a 9% decrement from peak AC throughout the competitive season (Table 1). The decrease was despite the significant increase in training load incurred by competitive season game play and training sessions. Ideally, players should be able to sustain peak performance capacity throughout the competitive season despite the accumulated training stress. While there was no significant change in AC throughout the competitive season (B4–B5), our sample competed with a 9% (*p* = 0.01) disadvantage from their peak AC potential observed at the conclusion of the spring season (B3) which has previously been shown to negatively impact athletic performance [2,15,16,17]. The sustained reduction in performance (AC) beyond two weeks is an indicator of overtraining [4,8,24,25], which our population experienced. Furthermore, female team sport athletes are 26% more likely to become overtrained [27] and therefore planned rest intervals throughout the competitive season is likely to benefit female athletes’ performance [18,19,20].

When considering the fluctuation of AC throughout the ATC, no significant differences in AC were observed when comparing B2 and B4 time points (after the transition period (B2) and prior to the competitive season (B4)) (Figure 1 and Figure 2). The negative net AC value (3.2%) across this timeline suggests that six months of preparatory training prior to the competitive season was unsuccessful. Prior to B2, there was a 10 wk transition period where training load was significantly reduced and AC increased (Table 1; Figure 2). Throughout B2–B4, training load did increase in an effort to prepare for the competitive season that consisted of both directed and indirect training. However, our data show that AC was no different at the start of the competitive season (B4) than after nine weeks of rest despite the increase in training load in a directed coaching style from B2 to B3. Interestingly, despite lack of differences in AC at B2 and B4 time points, AC peaked at B3 and then subsequently dropped by ~7% (*p* = 0.021) (Table 1; Figure 2). These findings further illustrate the variation in effectiveness of the direct/indirect coaching strategies throughout these time blocks. Moreover, the reduced AC observed in the pre-season (B3–B4) remained suppressed throughout the competitive season (B4–B5) regardless of the directed coaching style of the competitive season (B4–B5). This suggests that if AC is suppressed approaching the competitive season, it will remain so regardless of coaching style until rest is implemented within the training regimen. Lastly, physiological adaptations are elastic in that they can be lost or minimized if coaches do not appropriately plan their strategies and can positively and/or negatively affect their players.

Fat-free mass is known as the principle tissue responsible for oxygen utilization during exercise and therefore players with a higher FFM typically possess a higher AC [15,26]. While current research with elite soccer athletes suggests that FFM has a positive effect on AC [2,15,26], our results suggest that there is no relationship between FFM and AC throughout the transition period (B1–B2) where rest was prioritized (Figure 2). A possible explanation for this response is that our population experienced a significant reduction in training load (volume + intensity) following the 2016 competitive season (B1) with no scheduled workouts and minimal communication with coaches. Additionally, it is important to note that our results contradict previous literature with respect to FFM influencing AC [4,14,23]. Therefore, adequate rest, which the athletes in this study experienced within the transition period, could potentially mitigate accumulated stress incurred as a result of the heavy training loads facilitated during the competitive season [5]. Coaches should consider strategic rest period planning to help balance and accommodate the large training loads incurred during the preparatory and competitive seasons.

A limitation to the current study is the lack of menstrual cycle status and contraceptive practices among our population at each testing block throughout the ATC. Sex hormones are considered to have a mild to low effect on performance in eumenorrheic women [37,38]. However, a recent systematic meta-analysis of the effect of the menstrual cycle on performance shows that relevant literature lacking continuity and rigor. Furthermore, the literature suffer from low to moderate effect sizes, making it difficult to discern the impact the menstrual cycle has on performance among high-level female athletes [37].

## 5. Conclusions

Our results indicate that coaching strategy, training style, training load, and rest affect both AC and FFM throughout seasonal transitions within the ATC. Large increases in training volume along with insufficient recovery may result in accumulated physiological stress, which is shown to negatively affect AC and body composition in collegiate athletes. Therefore, regular testing is warranted to evaluate the physiological status of collegiate athletes throughout the ATC. Alternatively, rest was shown to positively impact AC in our population, as did CT coupled with rest throughout the spring season (B2–B3). The combined approach that included CT paired with adequate recovery provided the greatest improvement in AC and FFM (B1–B3). However, the indirect coaching style could have led to a perceived lack of coaching support during the pre-season (B3–B4) that negatively influenced training motivation and adherence, thus reducing AC. The impact of a reduced AC that occurred after the pre-season (B3–B4) had residual effects across the competitive season (B4–B6) that our sample could not overcome. The large training loads that include high volume and intensity paired with little recovery throughout the competitive season prevented positive training adaptations despite the physiological stimulus to do so. Therefore, pre-planning training outcomes with directed training and programming alongside purposeful rest and recovery strategies throughout the annual training cycle is ideal to maintain peak physiological status. Additionally, regular testing is warranted to monitor AC and body composition, as they can be used as indicators of training stress accumulation and performance changes in Division 1 female soccer players.

## Figures and Tables

**Figure 1 sports-08-00107-f001:**
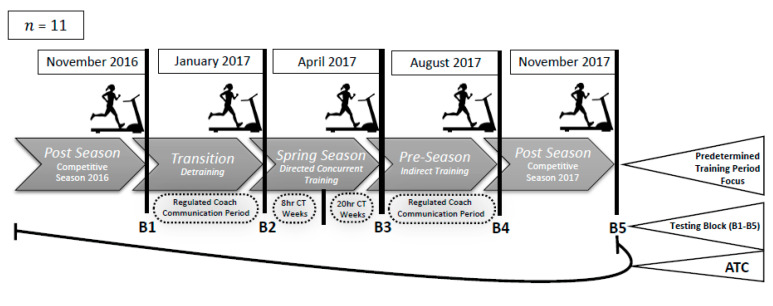
Exercise design includes maximal aerobic capacity (AC) and body composition testing across all predesignated time blocks (B1–B5) throughout the annual training cycle (ATC). Seasonal transitions and training period focus are identified intermittently between testing blocks. Regulated coach communication periods are mandated by the Division I National Collegiate Athletics Association compliance rulebook [29]. (CT) designates when concurrent training style was implemented during designated time blocks.

**Figure 2 sports-08-00107-f002:**
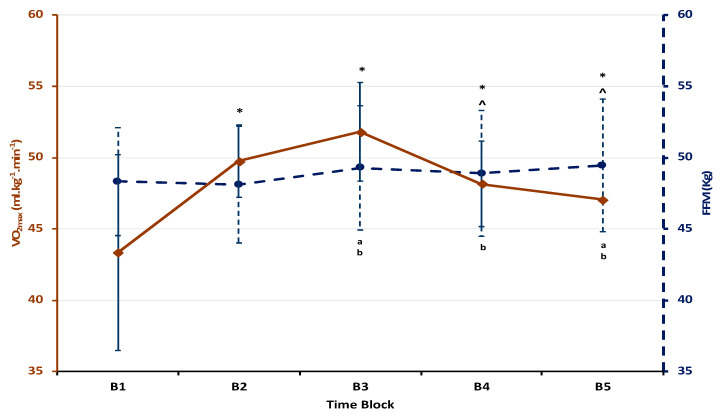
Seasonal variation of maximal aerobic capacity (solid line) and fat-free mass (FFM) (dotted line) across the annual training cycle (ATC). Data are presented as the mean + SD. * Aerobic capacity (AC) significantly different from B1 (*p* < 0.01). ^ Aerobic capacity (AC) significantly different from B3 (*p* < 0.01). (a) Fat-free mass (FFM) significantly different from B1 (*p* < 0.05). (b) Fat-free mass (FFM) significantly different from B2 (*p* < 0.05).

**Table 1 sports-08-00107-t001:** Pairwise comparisons of maximal aerobic capacity (AC) across all predesignated time blocks represented within the annual training cycle (ATC). Data are presented as the mean + SD, percent difference (%diff) and effect size (ES) with each time block (B1–B5) comparison.

Block	Aerobic Capacity (ml·kg^−1^·min^−1^)		(B1) Post-Season 2016		(B2) Transition		(B3) Spring-Season		(B4) Pre-Season		(B5) Post-Season 2017	
	Mean	SD	% Diff	ES	% Diff	ES	% Diff	ES	% Diff	ES	% Diff	ES
B1	43.34	3.34	-	-	−12.90%	0.50	−16.35%	0.81	−10.00%	0.57	−8.14%	0.50
B2	49.75 *	7.20	12.90%	0.50	-	-	−3.98%	0.19	3.20%	0.14	5.27%	0.24
B3	51.81 *	2.67	16.35%	0.81	3.98%	0.19	-	-	7.04%	0.50	8.93%	0.78
B4	48.16 *^	3.62	10.00%	0.57	−3.20%	0.14	−7.04%	0.50	-	-	2.03%	0.17
B5	47.03 *^	3.14	8.14%	0.50	−5.17%	0.24	−8.93%	0.78	−2.03%	0.17	-	-

(*) designates significant difference from B1 (*p* < 0.01). (^) designates significant difference from B3 (*p* < 0.01).

**Table 2 sports-08-00107-t002:** Pairwise comparisons of fat-free mass (FFM) across all predesignated time blocks represented within the annual training cycle (ATC). Data are presented as the mean + SD and percent difference (%diff), and effect size (ES) with each time block (B1–B5) comparison.

Block	FFM (kg)		(B1) Post-Season 2016		(B2) Transition		(B3) Spring-Season		(B4) Pre-Season		(B5) Post-Season 2017	
	Mean	SD	% Diff	ES	% Diff	ES	% Diff	ES	% Diff	ES	% Diff	ES
B1	48.31	3.96	-	-	−0.44%	0.50	1.98%	0.81	1.19%	0.57	2.31%	0.50
B2	48.10	4.29	0.44%	0.03	-	-	2.40%	0.19	0.81%	0.14	2.74%	0.24
B3	49.28 *^	4.57	−2.02%	0.11	−2.46%	0.19	-	-	−0.80%	0.50	0.36%	0.78
B4	48.89 ^	4.62	−1.21%	0.07	−1.65%	0.14	0.79%	0.50	-	-	1.13%	0.17
B5	49.45 ^	4.87	−2.37%	0.13	−2.82%	0.24	−0.34%	0.78	1.68%	0.17	-	-

(*) designates significant difference from B1 (*p* < 0.01). (^) designates significant difference from B3 (*p* < 0.01).

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
