# Peer review of "A Longitudinal Prospective Study: The Effect of Annual Seasonal Transition and Coaching Influence on Aerobic Capacity and Body Composition in Division I Female Soccer Players"

_sports, 2020, doi:10.3390/sports8080107_

Round 1

Reviewer 1 Report

The work was aimed at assessment the effect of seasonal transition, training stress, and coaching influence throughout the annual training cycle on aerobic capacity (AC) with consideration of body composition in Division I female soccer players. The study is very interesting especially that there is little number of research conducted on women. It is well designed and clearly presented. However, I have some comments I’d like to express.

  1. There is a lack of participant’s hormone status data regarding the menstrual cycle. It might have some influence on the AC results and it would be interesting to evaluate it. Perhaps the Authors have some literature data on this topic evidencing the influence or a that there is no influence of menstrual cycle status and physiological parameters / body composition variables?
  2. The Authors state that “During the transition period, the NCAA regulates (limits) coach communication with athletes [32]. The combined effect of no training prescription and limited communication produced (…)” (lines 233-234). I am not familiar with NCAA regulations. However, I was wondering if the coach did not (or cannot) prescribe detailed training program for the athletes for transition period? Do the participants train on their own during this period? If the coach can (and do) leave the participants training program, it should be constructed to manage participant’s aerobic capacity.
  3. I believe there is a mistake in numbering the Figures. After figure 1 there is a Figure 4 in the manuscript and the Authors refer only to Figures 1 and 2.
  4. A few typographic mistakes (incorrect spaces) should be corrected (e.g. lines 69, 71).

Author Response

The work was aimed at assessment the effect of seasonal transition, training stress, and coaching influence throughout the annual training cycle on aerobic capacity (AC) with consideration of body composition in Division I female soccer players. The study is very interesting especially that there is little number of research conducted on women. It is well designed and clearly presented. However, I have some comments I’d like to express.

THANK YOU FOR YOUR KIND COMMENTS AND INSIGHT.

  1. There is a lack of participant’s hormone status data regarding the menstrual cycle. It might have some influence on the AC results and it would be interesting to evaluate it. Perhaps the Authors have some literature data on this topic evidencing the influence or a that there is no influence of menstrual cycle status and physiological parameters / body composition variables?

THANK YOU FOR YOUR COMMENTS. SEX BASED HORMONES ARE AN IMPORTANT CONSIDERATION AND THEREFORE WE HAVE ADDED RELEVANT LITERATURE AND DISCUSSION REGARDING THE HORMONE STATUS AT THE END OF THE DISCUSSION AS A LIMITATION.

“A limitation to the current study is the lack of menstrual cycle status and contraceptive practices amongst or population at each testing block throughout the ATC. Sex hormones are considered to have a mild to low effect on performance in eumenorrheic women [36,37]. However, a recent systematic meta-analysis of menstrual cycle effect on performance show the relevant literature lacking continuity and rigor. Furthermore, the literature suffer from low to moderate effect sizes making it difficult to discern the impact menstrual cycle has on performance amongst high level female athletes [36].”

  1. The Authors state that “During the transition period, the NCAA regulates (limits) coach communication with athletes [32]. The combined effect of no training prescription and limited communication produced (…)” (lines 233-234). I am not familiar with NCAA regulations. However, I was wondering if the coach did not (or cannot) prescribe detailed training program for the athletes for transition period? Do the participants train on their own during this period? If the coach can (and do) leave the participants training program, it should be constructed to manage participant’s aerobic capacity.

THE TRANSITION PERIOD WAS A TIME WHERE NO EXERCISE PRESCRIPTION WAS GIVEN TO THE ATHLETES. EXERCISE PRESCRIPTIONS CAN BE GIVEN TO THE ATHLETES IN LINE WITH NCAA REGULATIONS AT THE COACHES DESCRETION. HOWEVER, THE ATHLETES ARE NOT REQUIRED TO COMPLETE IT. WITH OUR POPULATION, THE ATHLETES WERE FREE TO EXERCISE AT WILL. HOWEVER, NO SPECIFIC DIRECTION WAS GIVEN DURING THIS TIMEFRAME. FURTHERMORE, THE NCAA REGULATES COMMUNICATION BETWEEN THE ATHLETES AND COACHES. WE ADDRESS THIS WITHIN THE METHODS SECTION:

“The transition period (B1-B2) totaled 10 weeks and immediately followed the 2016 competative season. Moreover, the transition period included an indirect training method with no exercise prescription and a NCAA regulated/limited coaching communication period [27].”

  1. I believe there is a mistake in numbering the Figures. After figure 1 there is a Figure 4 in the manuscript and the Authors refer only to Figures 1 and 2.

THANK YOU FOR ADDRESSING THIS OVERSIGHT. THE TERM “Figure 4” HAS BEEN REMOVED FROM THE MANUSCRIPT AND REPLACED WITH “Figure 2”  AS REFERENCED WITHIN THE MANUSCRIPT TEXT.

  1. A few typographic mistakes (incorrect spaces) should be corrected (e.g. lines 69, 71).

INCORRECT SPACING HAS BEEN REVISED.

Reviewer 2 Report

Overall this is a well written paper with an interesting result on sport science area. The results are based on rational working hypothesis and it is admirable the quantity of variables used. However, the sample is too short to obtain conclusions.
INTRODUCTION
The introduction provides enough background information for readers to understand the problem. The introduction provides a good perspective of the main topic.
Motivations for this study are more than clear. The objectives are clearly defined at the Introduction, the argumentation in this last part was concise and clarifying.
METHODS
The experimental apparatus or equipment are quite standard. However, they are appropriate for the aim of the study.
More information about the specific training developed would be interesting. In addition, more information about the sample is neccesary. However, it is more than sufficient to reproduce the research.
RESULTS
Results paragraph should include more relevant and extended data. Seems to be lacking some explaining information.
All of the tables include specific, good developed statistic. 
DISCUSSION
All possible interpretations of the data considered are consistent.
Limitations are well established and comprehensive.
The conclusions have coherence with the initial hypothesis, in addition, they are well established and according to the present discussion.
LITERATURE CITED
The literature cited is relevant to the study

SIGNIFICANCE AND NOVELTY
As it stands, the results are novel and important enough for this journal.   MINORS:  Abstract: Describe ATC abbreviation, please  

Author Response

Overall this is a well written paper with an interesting result on sport science area. The results are based on rational working hypothesis and it is admirable the quantity of variables used. However, the sample is too short to obtain conclusions. 

THANK YOU FOR YOU KIND ASSESSMENT OF OUR SUBMISSION. IN REFERENCE TO YOUR COMMENTS REGARDING THE STUDY SAMPLE SIZE, WE HAVE ADDRESSED THE SAMPLE SIZE WITHIN THE METHODS SECTION, SPECIFICALLY “Exercise Design” WITH THE STATEMENT, “Sample size was determined using an a priori power analysis with an effect size of 0.68 from Miller et al. [2]and a power of 0.98.”

INTRODUCTION
The introduction provides enough background information for readers to understand the problem. The introduction provides a good perspective of the main topic.
Motivations for this study are more than clear. The objectives are clearly defined at the Introduction, the argumentation in this last part was concise and clarifying.

THANK YOU FOR THE KIND ACCOLADES REGARDING THE INTRODUCTION/BACKGROUND.

METHODS
The experimental apparatus or equipment are quite standard. However, they are appropriate for the aim of the study. More information about the specific training developed would be interesting. In addition, more information about the sample is neccesary. However, it is more than sufficient to reproduce the research.

THE INFORMATION REGARDING THE TRAINING STYLE IS OUTLINED WITHIN THE METHODS SECTION, SPECIFICALLY “Specified Training Season Focus”. HOWEVER, REVISED LANGUAGE TO ENHANCE THE TRAINING PERIOD DESCRIPTIONS RELATED TO THE DIRECT AND INDIRECT STYLES ARE INCLUDED:

“Each testing block followed periodized training periods with pre-determined team training goals implemented by the coaching staff depicted in Figure 1. The transition period (B1-B2) totaled 10 weeks and immediately followed the 2016 competative season. Moreover, the transition period included an indirect training method (no direct coach oversight) with no specified exercise prescription and a NCAA regulated/limited coaching communication period [27]. The 12 week spring season (B2-B3) increased training volume, frequency, and intensity which included regular bouts of directed (strength coach facilitated) CT (>3x week) along with regular aerobic and velocity training that had a sport specific focus. Fifty percent of the spring season consisted of an 8hr/wk CT period and the remaining 50% 20hr CT weeks. The preseason (B3-B4) was 15 weeks long and employed an indirect training method (no direct coach oversight) where subjects were given a periodized CT program with specified aerobic, anaerobic, and strength-based exercises that included frequency, volume, and intensity prescriptions. Throughout the preseason, subjects participated separately in “summer league play” with periodic competitions that varied sample wide. The preseason was also a NCAA mandated period of limited coach communication [27] and therefore the potential for sample wide variation in training volume, intensity, and frequency was high. Lastly, the competative season (B4-B5) spanned 15 weeks, included a minimum of two NCAA collegiate competitions per week and ~four days of CT training/practice per week.”

RESULTS
Results paragraph should include more relevant and extended data. Seems to be lacking some explaining information.
All of the tables include specific, good developed statistic. 

INCLUDED WITHIN THE RESULTS SECTION ARE DESCRIPTIONS OF THE DATA ANALYSIS WITH TABLES AND FIGURES THAT INCLUDE RELEVANT STATISTICAL DATA. THE METRICS INCLUDE: MULTIVARIATE ANALYSIS, P VALUES, EFFECT SIZES, CONFIDENCE INTERVALS, AND PERCENT CHANGE. THE DATA IS PRESENTED IN WRITTEN FORM AND VISUALLY WITH BOTH TABLES AND FIGURES. IF THERE IS ANYTHING SPECIFIC THAT CAN IMPROVE THE ANALYSES WE ARE OPEN TO ANY AND ALL SUGGESTIONS ON BEHALF OF THE REVIEWER(S).

DISCUSSION
All possible interpretations of the data considered are consistent.
Limitations are well established and comprehensive.
The conclusions have coherence with the initial hypothesis, in addition, they are well established and according to the present discussion.

THANK YOU FOR YOUR KIND COMMENTS AND INSIGHT.

LITERATURE CITED
The literature cited is relevant to the study

SIGNIFICANCE AND NOVELTY
As it stands, the results are novel and important enough for this journal.   MINORS:  Abstract: Describe ATC abbreviation, please  

THE ATC ABBREVIATION IS DESCRIBED WITHIN THE ABSTRACT AS: “This study assessed seasonal transitions and coaching influence effect aerobic capacity (AC) and body composition across the annual training cycle (ATC).”

Reviewer 3 Report

This paper aimed to the aim of this study was to evaluate the effect of seasonal transition, training stress, and coaching influence throughout the ATC on AC with consideration of body composition in Division I female soccer players. I want to congratulate authors for the work done, however, some edits should be included to improve the quality of the manuscript.

The major concerns are related to the literature. As such, the following references should be included in introduction and discussion sections:

Buchheit M, Mendez-Villanueva A, Simpson BM, Bourdon PC. Match running performance and fitness in youth soccer. Int J Sports Med. 2010;31(11):818–25.

Castillo D, Los Arco Arcos A, Martínez-Santos R. Aerobic endurance performance does not determine the professional career of elite youth soccer players. Journal of Sports Medicine and Physical Fitness. 2018;58(4):392–8.

Tonnessen E, Hem E, Leirstein S, Haugen T, Seiler S. Maximal Aerobic Power Characteristics of Male Professional Soccer Players, 1989-2012. International Journal of Sports Physiology and Performance. 2013;8(3):323–9.

In addition, additional minor changes should be done:

Abstract

Explain FFM abbreviature.

Methods

I wonder about the reasons to use the aerobic capacity test in soccer players. Why did not authors perform other tests which resemble the players’ activity during match-play? For instance, the 30-15 IFT or YoYo IR1?

I encourage authors to better explain the aerobic capacity test. Explanation that could other researchers to implement this test protocol.

Authors state that “A 4x5 repeated measures ANOVA was used to analyze dependent variables: AC (ml•kg-1•min-171 1), FFM (kg), %BF (%), BM (kg) across five time blocks (B1-B5)”. However, repeated measures ANOVA was used to analyze the variations in each variable. Change, please.

Include formula to calculate percent change (%D),

Authors used the effect sizes (ES) to analyze the magnitude of differences among blocks.  Please, include the thresholds used of ES.

 Explain this abbreviature: LSD.

Author Response

This paper aimed to the aim of this study was to evaluate the effect of seasonal transition, training stress, and coaching influence throughout the ATC on AC with consideration of body composition in Division I female soccer players. I want to congratulate authors for the work done, however, some edits should be included to improve the quality of the manuscript.

The major concerns are related to the literature. As such, the following references should be included in introduction and discussion sections:

Buchheit M, Mendez-Villanueva A, Simpson BM, Bourdon PC. Match running performance and fitness in youth soccer. Int J Sports Med. 2010;31(11):818–25.

Castillo D, Los Arco Arcos A, Martínez-Santos R. Aerobic endurance performance does not determine the professional career of elite youth soccer players. Journal of Sports Medicine and Physical Fitness. 2018;58(4):392–8.

Tonnessen E, Hem E, Leirstein S, Haugen T, Seiler S. Maximal Aerobic Power Characteristics of Male Professional Soccer Players, 1989-2012. International Journal of Sports Physiology and Performance. 2013;8(3):323–9.

ALL SUGGESTED CITATIONS ARE NOW INCLUDED WITHIN THE MANUSCRIPT AND INCORPORATED THROUGHOUT BOTH THE INTRODUCTION/BACKGROUND AND DISCUSSION WHERE RELEVANT.

In addition, additional minor changes should be done:

Abstract

Explain FFM abbreviature.

THE FFM ABBREVIATURE EXPLAINATION HAS BEEN ADDED, “Height, weight, and body composition (fat free mass (FFM)) were measured. . .”

Methods

I wonder about the reasons to use the aerobic capacity test in soccer players. Why did not authors perform other tests which resemble the players’ activity during match-play? For instance, the 30-15 IFT or YoYo IR1?

THANK YOU FOR YOUR INSIGHT. AS I AM SURE YOU ARE AWARE, VO2MAX IS A LABORATORY TEST THAT IS CONSIDERED THE GOLD STANDARD FOR FITNESS ASSESSMENT. WHILE THE AUTHORS AGREE THAT SPORT SPECIFIC TESTING IS DIRECTLY APPLICABLE TO APPLIED FITNESS ASSESSMENT, THE YOYO IR1 AND 30-15 IFT USE PREDICTIVE EQUATIONS TO ASSESS PEAK PHYSIOLOGIC POTENTIAL (WITH ADDITIONAL METRICS FOR RECOVERY/CAPACITY, POWER, ETC) AND THEREFORE INCREASES POTENTIAL FOR VARIANCE IN TESTING MEASUREMENT. FURTHER, D1 COLLEGIATE ATHLETES ARE EXPECTED TO PERFORM AT OPTIMAL LEVELS AND MAINTAIN PEAK FITNESS. THE LABORATORY TESTING METHODOLOGY EMPLOYED THROUGHOUT THE CURRENT STUDY WAS SELECTED TO ACCURATELY ASSESS THE ATC AND COACHING INFLUENCE ON PEAK PHYSIOLOGICAL CAPACITY RATHER THAN SPORT APPLICABILITY. ADDITIONALLY, CITATIONS WITHIN THE BACKGROUND LITERATURE SHOW THAT VO2MAX IS A COMMON ASSESSMENT OF PERFORMANCE WITHIN SOCCER (MILLER ET AL.; ALVES ET AL.; PEART ET AL.).

I encourage authors to better explain the aerobic capacity test. Explanation that could other researchers to implement this test protocol.

THE AUTHORS HAVE FURTHER EXPANDED ON GXT PROTOCOL DESIGN FOR TESTING REPLICATION. THE LANGUAGE INCLUDED WAS CHANGED TO: “Each treadmill GXT protocol was designed to begin with a brisk walk (1.6-1.8 m•S-1). Each 1min stage increased velocity by 0.22-0.44 m•S-1 to ensure that volitional fatigue would occur within 8-12min [30,31]. Standardized criteria for the determination of VO2max along with a 15sec running average as described by Robergs et al. 2010 [32] were used to determine VO2max.”

Authors state that “A 4x5 repeated measures ANOVA was used to analyze dependent variables: AC (ml•kg-1•min-171 1), FFM (kg), %BF (%), BM (kg) across five time blocks (B1-B5)”. However, repeated measures ANOVA was used to analyze the variations in each variable. Change, please.

THE REPEATED MEASURES EXPERIMENTAL DESIGN DICTATED THE STATISTICAL ANALYSIS METHODOLOGY AND IS INDICATED WITHIN THE LITERATURE. IF THE REVIEWER HAS ANY SPECIFIC SUGGESTIONS THE AUTHORS ARE OPEN TO CONSIDERATIONS.

Include formula to calculate percent change (%D),

THE CALCULATION FOR PERCENT CHANGE WAS ADDED TO THE STATISTICAL ANALYSIS SECTION: “. . . %Δ=((final-initial/initial)*100). .” WITHIN THE METHODS SECTION.

Authors used the effect sizes (ES) to analyze the magnitude of differences among blocks.  Please, include the thresholds used of ES.

THE EFFECT SIZES THRESHOLDS WERE ADDED TO THE MANUSCRIPT AS: “. . .and pre-designated ES range limits were established as: low effect = 0.20-0.49; medium effect = 0.50-0.79; large effect 0.80-1.0.” 

Explain this abbreviature: LSD.

THE ABBREVIATION OF LSD HAS BEEN INCLUDED WITHIN THE STATISTICAL ANALYSIS SECTION: “When significant differences were observed, the Fisher’s Least Significant Difference (LSD) post hoc test. . .”